# Rapidly Determine the Maximum Power Point in the Parallel Configuration of the Photovoltaic System

**DOI:** 10.3390/s23177503

**Published:** 2023-08-29

**Authors:** Bui Van Hien, Truong Viet Anh, Nguyen Tung Linh, Pham Quoc Khanh

**Affiliations:** 1Faculty of Mechanical-Electrical and Computer Engineering, School of Technology, Van Lang University, Ho Chi Minh City 700000, Vietnam; hien.bv@vlu.edu.vn; 2Faculty of Electrical and Electronics Engineering, HCMC University of Technology and Education, Ho Chi Minh City 700000, Vietnam; anhtv@hcmute.edu.vn; 3Faculty of Control and Automation, Electric Power University, Hanoi 100000, Vietnam; linhnt@epu.edu.vn; 4Faculty of Electrical Engineering Technology, Industrial University of Ho Chi Minh City (IUH), Ho Chi Minh City 700000, Vietnam

**Keywords:** modified P&O algorithm, parallel PV configuration, fill factor, MPPT

## Abstract

The maximum power point tracking (MPPT) solutions improve power generation efficiency, quickly stabilizing the output waveform of photovoltaic (PV) systems under variable operating conditions. Along with new algorithms, improved and adjusted methods to exploit energy from PV systems are increasingly being researched and proposed. However, the proposed solutions based on the traditional algorithms and their improvements have poor performance, while the advanced algorithms or hybrid methods bring high performance but need to be simplified, and the response speed is higher. Moreover, a suitable PV configuration makes choosing a simple but highly efficient algorithm, especially in low-power PV system applications such as rooftop solar power, traffic lights, and moving vehicles…where the number of PV panels is insufficient to implement flexible configurations. This paper proposes a modified version of the Perturb and Observe (MPO) algorithm to improve MPPT performance and increase convergence speed in the parallel structure of PV panels. The Short-Circuit Current (I_sc_) and Open-Circuit Voltage (V_oc_) are calculated directly at specific operating conditions to quickly determine the potential maximum power point (MPP) that will reduce power interruptions and increase power generation efficiency compared to periodic updates. Therefore, the proposed solution converges faster, with higher efficiency, and the output signal in static and dynamic MPPT situations is more stable. The results show that the highest efficiency in simulation and experiment is 99.99% and 99.93%, respectively, while the convergence speed is 0.01 s and 0.03 s, respectively. They are better than the traditional Perturb and Observe (P&O) algorithm, the Variable Step Size Perturb and Observe (VSSP&O) method, and the Particle Swarm Optimization (PSO) technique under the same operating conditions. In addition, its performance and convergence speed are also compared with the latest introduced algorithms. The results show that it is valuable and reliable for parallel PV configuration.

## 1. Introduction

Rooftop photovoltaic systems are essential for apartments-integrated centralized generation, especially in high mountains or islands. Their advantages can be considered in some critical points as follows [1]: First, utilizing available space can generate electricity near where needed, reducing transmission and distribution losses associated with traditional centralized power generation. Secondly, they contribute to sustainable energy development by reducing reliance on conventional fossil fuel-based electricity, significantly reducing greenhouse gas emissions and environmental impact. Finally, they can result in significant cost savings over time due to low maintenance requirements and can generate electricity for a long time. Additionally, depending on local regulations and policies, excess electricity can be fed back into the grid.

However, in low-power PV system applications, due to the limitation of PV quantity, it is less efficient to implement complex configurations such as total-cross-tied, bridge-link, honeycomb, etc… Meanwhile, the parallel structure has outstanding advantages such as high power generation efficiency, low loss, and less variable I(V) and P(V) characteristic curves. Even under partial shading conditions, the observed characteristics consistently display a singular peak, and the maximum power remains unaffected by the bypass diodes present in this configuration [2]. Therefore, applying MPPT algorithms to efficiently improve the performance and convergence speed and quickly stabilize under the environment changes continuously is simple.

Further, along with the continuous development of PV systems globally due to the rapid energy demand growth, solutions to determine the MPP of PV systems are also proposed, adjusted, and improved continuously. They can be classified into conventional, intelligent, optimization, and hybrid methods [3,4]. The classical algorithms are simple and low-cost, but the convergence speed and efficiency must improve.

In contrast, advanced methods have high efficiency and speed but are complicated and expensive. As a result, hybrid or improved methods are proposed to optimize the strengths of individual algorithms. However, finding a solution that achieves high efficiency, fast tracking speed, low cost, and less complexity is challenging.

To solve the above problem, many studies have been introduced in different approaches and treatments. Recent publications concentrate on the traditional techniques combined with the I(V) characteristic curve of PV systems to improve efficiency and speed due to simplicity and low cost. By using the FF value to determine the power loss between the MPP under standard conditions and when shaded, document [5] has provided a solution to continuously compare actual working parameters with standard conditions to adjust the working position. While the authors in [6,7,8] proposed a solution to approximate the current at MPP (I_mp_ = k_i_I_sc_) to improve MPPT efficiency, convergence rate, and output stability. The PV module’s I_sc_ value is measured by interrupting the system’s regular operation with a particular frequency, storing the calculated value. However, speed and performance have yet to be significantly improved. Further, some studies focus on the V_oc_ value to determine MPP (V_mp_ = k_v_V_oc_) [8,9,10].

In the same way, the V_oc_ is also achieved by disconnecting the regular operation of the system. The significant advantages of these approaches are simple and low price. Yet, its drawback is that the interrupted system operation yields power losses when scanning the entire control range. If the sampling period of these two parameters is small, the accuracy of MPPT will increase, but it also increases the supply interruption time between source and load. Conversely, if they are not regularly updated, the calculated values may not be the optimal MPP of the PV system because I_sc_ and V_oc_ values are affected by operating conditions and aging.

To overcome this drawback, in the reference [11], the authors introduced an enhanced approach to improve the conventional P&O and INC methods. This method aims to optimize the maximum output power of the PV system and identify the most suitable design variable for regulating the step size in conventional algorithms. This solution can achieve a convergence speed of 0.0434 s and an efficiency of 99.07%. In [12], the Low Burden Narrow Search (LBNS), a reduced search space exploration metaheuristic algorithm for MPPT, is proposed. The main goal of this solution is to confine the search space around the actual optimum to minimize the number of update equations needed, effectively avoiding the exploration of irrelevant regions and decreasing computational steps. The simulation efficiency of this solution is 99.98% at 0.038 s. The document [13] introduces a variety of intelligent algorithms, such as Modified Incremental Conductance (MIC), the Cuckoo Search Algorithm (CSA), Gray Wolf Optimization (GWO), and PSO. Their efficiency ranges from 78.42% to 99.99% depending on operating conditions, and the fastest convergence speed is 0.14 s. The convergence speed is slightly improved in [14], about 0.0375 s, but its performance is only 99.54%. This paper presents a novel GMPPT algorithm that combines the Measurement Cells (MC) algorithm and the P&O method. Firstly, it utilizes the fast dynamic MC algorithm to identify and evaluate the local maximum power points. Then, the P&O method is employed to approximate and adjust the voltage to approach the MPP determined in the previous step. Overall, the proposed GMPPT algorithm offers an innovative approach by combining the strengths of the MC algorithm and the P&O method to efficiently track and maintain the maximum power point of the photovoltaic system. Most recently, the [15] document mentions an entirely new algorithm. The authors introduced the musical chairs algorithm (MCA), inspired by the game of musical chairs. Its outstanding advantage is that there is only one tuning parameter and variable swarm size, which makes it much easier to tune than other algorithms. Then, the adjusted version (BMCA) introduced in [16] can reduce error and convergence speed by 40% compared to different optimization algorithms while maintaining a zero failure rate. Therefore, the convergence speed of the MCA of 0.3 s has been improved up to 0.096 s in the BMCA version. Although it is more straightforward than other optimization algorithms, it is quite complex compared to traditional algorithms. In [17], a hybrid approach between the Fuzzy Logic Controller and the Cuckoo Optimization Algorithm (COA-FLC) to increase convergence speed, improve performance, and reduce oscillations around MPP is proposed. This combination increases the convergence speed of the solution to 0.016 s, and the efficiency is about 99.83%. A hybrid solution in [1] applies a hybrid MPPT control algorithm between PSO and P&O to ensure the system’s effectiveness even under various irradiance conditions. However, it only achieves simulation and experimental efficiency of about 92% and 90%, respectively.

The above analysis shows that there are many criteria to evaluate the effectiveness of an MPPT solution. However, have yet to find a solution that satisfies all evaluation criteria. This study introduces a simple MPPT method, low cost, high performance, and fast convergence speed, and can be widely applied in different operating conditions. The proposed solution is based on a modified P&O algorithm to quickly determine the potential MPP of the photovoltaic system. It is a simple, robust algorithm with high MPPT performance under standard conditions [4]. Further, the characteristic curves of parallel configuration always show the same under all operating conditions. Therefore, applying a modified P&O algorithm to this configuration will be helpful in low-voltage solar power applications due to its simplicity and low cost.

The outstanding contributions of this study include:Investigate some typical PV modules’ I(V) characteristics to determine the linear region where the I_sc_ value can be calculated directly. This data can be helpful for further research on photovoltaic systems;Suggest a cut-off point of 0.4 V_oc_ to directly calculate the I_sc_ value within this limit;Propose a method to directly determine I_sc_ and V_oc_ according to operating conditions using linear extrapolation from two random points.Quickly determine the duty cycle value at the potential MPP (d_mp_), which is then used as the starting point for the MPO algorithm.Improve convergence speed and MPPT performance of parallel PV panels under different operating conditions.

The simulation and experimental results show that the proposed method’s convergence speed and MPPT efficiency are better than other methods. It can effectively be applied in parallel PV systems with low voltage and power.

The paper is organized as follows: Section 2 presents the characteristics of PV systems under different operating conditions, and Section 3 details the principle of the Boost converter. The I(V) characteristic curves of some typical PV modules and methods to determine I_sc_, V_oc_ are presented in this section. In Section 4, simulation and experimental results are shown and discussed. Finally, Section 5 draws conclusions from the study.

## 2. Effects of Working Conditions on PV System

Equation (1) presents the relationship between the output current and voltage of a typical PV cell that is introduced in Figure 1 [18,19,20].
(1)$$I_{pv}=I_{ph}-I_{0}\left\{e^{\frac{q\left(V_{pv}+I_{pv}R_{s}\right)}{nKT_{c}}}-1\right\}-\frac{V_{pv}+I_{pv}R_{s}}{R_{sh}}$$
where V_pv_ and I_pv_ are the output voltage (V) and current (A) of PV, respectively; I_ph_ is the light current (A); I_0_ is the reverse saturation current of the diode (A); q is the electron charge (1.602 × 10^−19^ C); K is the Boltzmann constant (1.381 × 10^−23^ J/K); Tc is the cell temperature (K); R_s_, R_sh_ are series and shunt resistance, respectively (Ω); n is diode ideality factor (dimensionless).

In which the light current I_ph_ depends on the solar irradiation and the working temperature of the PV cell, as stated in (2) [20,21].
(2)$$I_{ph}=\left[I_{sc}+\alpha_{i}\left(T_{c}-T_{ref}\right)\right]\frac{W}{W_{r\mathrm{ef}}}$$
where I_sc_ is short circuit current (A); T_ref_ is the temperature at standard condition (K); W is the solar irradiance level (W/m^2^); W_ref_ is the irradiance level at standard state (W/m^2^); α_i_ is the temperature coefficient of I_sc_ (mA/°C).

At any operating condition, the component currents of the PV cell are introduced as in (3).
(3)$$I_{pv}=I_{sc}-I_{D}-I_{R_{sh}}$$
where I_D_ and I_Rsh_ are the diode-current and the shunt-resistance-current, respectively. Their values are determined according to Equations (4) and (5).
(4)$$I_{D}=I_{0}\left\{e^{\frac{q\left(V_{pv}+I_{pv}R_{s}\right)}{nKT_{c}}}-1\right\}$$
(5)$$I_{R_{sh}}=\frac{V_{pv}+I_{pv}R_{s}}{R_{sh}}$$

The graphs of Equations (3)–(5) are shown in Figure 2. In which the nonlinear region on the I(V) curve is caused by the I_D_ current. In contrast, the slope in this area is mainly due to the I_Rsh_ current. However, these operating zones are affected by radiation and temperature (Figure 3). Furthermore, under the same operating conditions, its slope depends on the R_sh_ value (Figure 4). Therefore, surveying the working regions of PV modules under different operating conditions will simplify calculating their parameters.

## 3. The Duty Cycle of the PV Control System

### 3.1. The Relationship between Duty Cycles of the Boost Converter

The boost converter in Figure 5 provides maximum power to the load. The MPPT block controls the DC/DC converter to keep the system working at the highest PV efficiency by adjusting the output voltage (V_out_) according to the input voltage (V_in_) through a duty cycle (d) value (0 < d < 1). The relationship between the input and output voltages is presented in (6) [22,23].
(6)$$\frac{V_{out}}{V_{in}}=\frac{1}{1-d}$$

If the losses on the DC/DC circuit are negligible, the V_pv_ and I_pv_ values are determined based on a random d value, and the relationships between them and resistive load (R_L_) can be introduced as in Equation (7).
(7)$$\frac{V_{pv}}{I_{pv}}=R_{L}{\left(1-d\right)}^{2}$$

In the range 0 < d < 1, only one duty cycle value achieves the maximum output power under the input power changes, and the load R_L_ remains constant. However, as the operating conditions change, the potential MPP (P_mp_ = V_mp_ × I_mp_) will also change. Therefore, the corresponding estimated duty cycle (d_mp_) must also be recalculated. The relationship between them is shown in (8) [23].
(8)$$\frac{V_{mp}}{I_{mp}}=R_{L}{\left(1-d_{mp}\right)}^{2}$$

Combine Equations (7) and (8) to determine d_mp_ at MPP, as shown in (9).
(9)$$d_{mp}=1-\left(1-d\right)\sqrt{\frac{V_{mp}}{I_{mp}}\frac{I_{pv}}{V_{pv}}}$$

The I_pv_ and V_pv_ values are measured and stored at any value of d. When the working environment changes, it is difficult to directly determine the d_mp_ value. However, the stability of the characteristic curve of the parallel configuration under all operating conditions makes it simpler to calculate the V_mp_ and Imp values according to FF. In other words, if the I_sc_ and V_oc_ values are known, the MPP(V_mp_, I_mp_) will be estimated following (10) [5,6,7,8,9,10], and then calculate d_mp_ following (9).
(10)$$V_{mp}=k_{v}V_{oc}\;\;\;\;and\;\;\;I_{mp}=\frac{FF}{k_{v}}I_{sc}=k_{i}I_{sc}$$

As analyzed above, if the I_sc_ and V_oc_ values are updated periodically, the solution will be less accurate because they are affected and changed continuously by the environment. On the contrary, when measured under actual conditions, it will increase convergence speed and reduce error. Therefore, it is necessary to investigate the different working conditions of PV modules to find a solution to quickly calculate these parameters.

### 3.2. The Relationship between Duty Cycles at the MPP

To propose a solution to quickly calculate Isc and V_oc_ in real time, in this study, radiation and temperature were adjusted from W_1_ = 200 W/m^2^ to W_2_ = 1000 W/m^2^ and T_1_ = 0 °C to T_2_ = 60 °C, respectively (Figure 3). At these limits, the resistance value of the PV system is from R_in_1_ at M_1_(W_1_, T_1_) to R_in_2_ at M_2_(W_2_, T_2_). The relationship between the parameters at these two points is shown in (11).
(11)$$\frac{V_{mp1}}{I_{mp1}{\left(1-d_{mp1}\right)}^{2}}=\frac{V_{mp2}}{I_{mp2}{\left(1-d_{mp2}\right)}^{2}}$$

Solve Equation (11) to get Equation (12) as follows.
(12)$$\frac{{\left(1-d_{mp2}\right)}^{2}}{{\left(1-d_{mp1}\right)}^{2}}=\frac{V_{mp2}I_{mp1}}{I_{mp2}V_{mp1}}$$

Combine Equations (10) and (12), the relationship between duty cycles at MPP (d_mp_), as shown in (13).
(13)$$\frac{{\left(1-d_{mp2}\right)}^{2}}{{\left(1-d_{mp1}\right)}^{2}}=\frac{k_{v}V_{oc2}k_{i}I_{sc1}}{k_{i}I_{sc2}k_{v}V_{oc1}}=\frac{V_{oc2}I_{sc1}}{I_{sc2}V_{oc1}}$$

Since M_1_ and M_2_ operate differently from the standard conditions, Equation (13) is rewritten as in (14). Where α_v_ is the temperature coefficient on V_oc_ (mV/°C), and α_i_ is the temperature coefficient of I_sc_ (mA/°C).
(14)$$\frac{{\left(1-d_{mp2}\right)}^{2}}{{\left(1-d_{mp1}\right)}^{2}}=\frac{\left(V_{oc}-\alpha_{v}(T_{2}-25)\right)}{W_{2}\left(I_{sc}+\alpha_{i}(T_{2}-25)\right)}\frac{W_{1}\left(I_{sc}+\alpha_{i}(T_{1}-25)\right)}{\left(V_{oc}-\alpha_{v}(T_{1}-25)\right)}$$

Substitute the data in Table 1 [24] into Equation (14) to calculate the relationship between duty cycles at M_1_ and M_2_, as shown in (15).
(15)$$\frac{{\left(1-d_{mp2}\right)}^{2}}{{\left(1-d_{mp1}\right)}^{2}}=1.65$$

Equation (15) shows that the relationship between d_mp1_ and d_mp2_ under the best and worst operating conditions is established based on the technical characteristics of the PV module. To reduce the pressure on the DC/DC converters switches, the d_mp_ values should be around 0.5 [25,26]. To satisfy all operating conditions within this survey limit, duty cycle values must be satisfied as in (16).
(16)$$0.5-d_{mp1}=d_{mp2}-0.5$$

Combine Equations (15) and (16) to get the d_mp1_ = 0.29 and d_mp2_ = 0.71.

In the same way, the survey results for the remaining PV modules [24] are also collected and detailed in Table 2.

### 3.3. The Linear Region on the I(V) Curve

The working areas of the I(V) curve in Figure 3 show that the d_mp2_ value (at M_2_) is suitable for calculating I_sc_ under all operating conditions. Because the resistance value of the PV system at 1000 W/m^2^—60 °C (R_in_2_) belongs to line OM_2_, it always intersects the linear region of the curves. In contrast, d_mp1_ was chosen as the reference limit to calculate the V_oc_ value of the PV system.

According to reference [27], the authors have proposed a working point A(0.2 V_oc_; I_sc_) on the I(V) curve to calculate I_sc_. That is, at a particular operating condition, the relationship between the parameters at A and MPP(V_mp_; I_mp_) is presented as in (17) after combining Equations (9) and (10).
(17)$$d_{A}=1-\left(1-d_{mp2}\right)\sqrt{\frac{0.2V_{oc}}{I_{sc}}\frac{0.93I_{sc}}{0.8V_{oc}}}=0.86$$

With d_mp2_ = 0.71, the limit for calculating the I_sc_ of PV module MSX-60. The values of k_v_ and k_i_ are chosen as 0.80 and 0.93, respectively, because they satisfy all PV modules surveyed in Table 2. The duty cycle at 0.2 V_oc_ is d_A_ = 0.86, which the I_sc_ can be calculated directly without interrupting the power supply for periodic updates.

However, the PV module’s R_sh_ value causes the characteristic curve slope in the linear region. Therefore, the calculation of the I_sc_ value at A (corresponding to R_sh1_) can be accurate, but the error increases significantly when calculating at A′ (corresponding to R_sh2_) (Figure 4). The larger this resistance fluctuates, the greater the calculation error. It causes a decrease in the MPPT efficiency of the PV system. The analysis shows that calculating I_sc_ at A(0.2 V_oc_, I_sc_) is less accurate in actual operating conditions. An extrapolation method is proposed from two points in the linear region to solve these problems. It is necessary to define the linear part on the I(V) curve corresponding to its d parameters. With a significant d value, the measured voltage and current are in the linear region but will pressure the DC/DC converter switches. On the contrary, the output signals may fall into the nonlinear zone, causing I_sc_ calculation error.

To analyze the active regions on the PV modules’ I(V) curve, their parameters are investigated according to V_oc_. If I_pv_R_s_ << V_pv_, the output voltage of the PV module will be considered in Equation (18) with 0 ≤ x ≤ 1.
V_pv_ = xV_oc_(18)

Then, Equations (4) and (5) are rewritten as in (19) and (20). Where k = 0.1 to 1.0 means survey shunt-resistor-current from 0.1 I_Rsh_ to I_Rsh_ to compare with I_D_.
(19)$$I_{D}=\left|I_{0}\left\{e^{\frac{qxV_{oc}}{nKT_{c}}}-1\right\}\right|$$
(20)$$I_{Rsh}=k\left|\frac{xV_{oc}}{R_{sh}}\right|$$

The 5-parameter model of the PV module MSX-60 is used to investigate according to Equations (19) and (20) under different temperatures and R_sh_. In which the I_pv_ is determined by Equation (3). The current waveforms in Figure 6 show that the linear region is limited in V_pv_ < 0.4 V_oc_. Then, the remaining PV modules were similarly surveyed. The survey results in Figure 7 show that their linear areas are in the voltage range of less than 0.4 V_oc_. In particular, the PV modules such as GxB-340 and Shell SQ150 have linear regions in V_pv_ < 0.5 V_oc_. Therefore, V_pv_ = 0.4 V_oc_ is the reference point for determining the d limit to calculate I_sc_ in this study.

However, output parameters are measured and stored based on the duty cycle value at specific times. It is necessary to determine the value of d at 0.4 V_oc_ of PV modules according to Equation (21).
(21)$$d_{A}=1-\left(1-d_{mp2}\right)\sqrt{\frac{0.4V_{oc}}{I_{sc}}\frac{0.93I_{sc}}{0.8V_{oc}}}=0.80$$

Equation (21) shows that the linear range is d ≥ 0.80 for PV module MSX-60. This value ensures that the measured current and voltage are always in the linear region of all operating conditions.

Similarly, the d_mp2_ values in Table 2 are used to calculate the duty cycle at 0.4 V_oc_ for the remaining PV modules. The results listed in Table 3 show that, within the survey range, the d value at 0.4 V_oc_ is from 0.72 to 0.80. Therefore, using any two points within this range to establish the equation of the line through I_sc_ will limit the disadvantage mentioned above. In this survey, d_1_ = 0.8 and d_2_ = 0.75 were chosen as the reference data to survey all PV modules.

**Figure 7 sensors-23-07503-f007:**
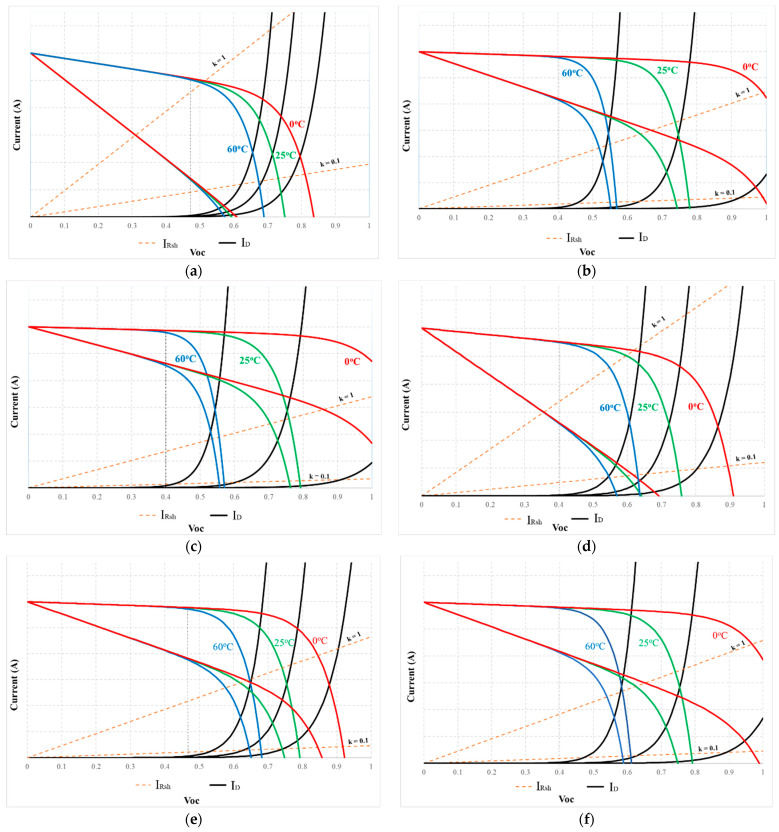
The I(V) characteristic curve of some typical PV modules: (**a**) GxB-340; (**b**) Shell ST40; (**c**) Shell S70; (**d**) SSt 230-60P; (**e**) Shell SQ150; and (**f**) Shell SP75.

### 3.4. Identifying the Value of the Isc

With two points [P_1_(V_1_, I_1_) and P_2_(V_2_, I_2_)] in the linear area of the I(V) curve corresponding to d_1_ và d_2_, the equation of the line through P_1_ and P_2_ is shown in (22).
(22)$$I=\frac{I_{2}-I_{1}}{V_{2}-V_{1}}V+\left[I_{1}-\frac{I_{2}-I_{1}}{V_{2}-V_{1}}V_{1}\right]$$

Accordingly, the value of I_sc_ is inferred by (23) when substituting V = 0 into (22) as follows.
(23)$$I_{sc}=I_{1}-\frac{I_{2}-I_{1}}{V_{2}-V_{1}}V_{1}$$

Applying Equation (23) calculates the I_sc_cal_ value of PV modules under different temperature and radiation conditions. The current deviation (ΔI%) between this result and the actual value (I_sc_) of PV modules is determined using Equation (24).
(24)$$\Delta I\%=\frac{\left|I_{sc}-I_{sc\_cal}\right|}{I_{sc}}100\%$$

The summary of the survey, calculation, and comparison results in Table 4 and Table 5 shows that the maximum deviation is 3.79% when operating conditions are 1000 W/m^2^ and 60 ^o^C (PV module Shell SQ150), and the minimum error is 0.00% in some cases. The maximum average deviation is 1.14% (PV module Shell SQ150), while the lowest difference is 0.04% (PV module SSt 230-60P). The results show that the proposed method can accurately estimate the I_sc_ value of the PV system.

### 3.5. Identifying the Value of the Voc

As analyzed above, the I_sc_ value is determined based on d_mp2_, while V_oc_ depends on d_mp1_ (Table 2). To calculate V_oc_ in all operating conditions, point B(V_B_, I_B_) must satisfy d_B_ ≤ d_mp1_, as shown in Equation (25).
(25)$$d_{B}=1-\left(1-d_{mp1}\right)\sqrt{\frac{0.93I_{sc}}{0.8V_{oc}}\frac{V_{B}}{I_{B}}}$$

In reference [20], the authors proposed calculating V_oc_ at I_B_ = 0.2 I_sc_. That is, the coordinates of point B(V_oc_, 0.2 I_sc_) are substituted into Equation (25) as follows.

In the worst operating condition (d_mp1_), the d_B_ value calculated from Equation (25) (PV MSX-60) is d_B_ = −0.71. In contrast, at best operating condition (d_mp2_), this value is 0.30. It shows that choosing V_oc_ = V_B_ at 0.2 I_sc_ cannot satisfy all operational requirements. The results of the same survey for the remaining PV modules are presented in Table 6.

The calculation results in Table 6 are suitable only within a narrow range of operating conditions. They do not represent PV modules under the operating conditions proposed in this study. Therefore, if V_oc_ is calculated at B, the error increases, and the convergence rate decreases significantly. Furthermore, if this value is calculated at d = 0, it will pressure the diode. In the same way, V_oc_ is also calculated from two points to limit the above defects. In this study, the values d_3_ = 0.1 and d_4_ = 0.05 were chosen as representatives for calculating V_oc_. The equation of the line passing through two points P_3_(V_3_, I_3_) and P_4_(V_4_, I_4_) is presented in Equation (26).
(26)$$I=\frac{I_{4}-I_{3}}{V_{4}-V_{3}}V+\left[I_{3}-\frac{I_{4}-I_{3}}{V_{4}-V_{3}}V_{3}\right]$$

Accordingly, the value of V_oc_ is inferred by (27) when substituting I = 0 into (26).
(27)$$V_{oc}=V_{3}-\frac{V_{4}-V_{3}}{I_{4}-I_{3}}I_{3}$$

Applying Equation (27) calculates the V_oc_cal_ value of PV modules under different temperature and radiation conditions. The voltage error (ΔV%) between this result and the actual value (V_oc_) of PV modules is determined using Equation (28).
(28)$$\Delta V\%=\frac{\left|V_{oc}-V_{oc\_cal}\right|}{V_{oc}}100\%$$

The summary results in Table 7 and Table 8 show that the most significant deviation is 4.69% (PV module Shell ST40) at 200 W/m^2^ and 25 °C. The minimum error is 0.02% (PV module MSX-60) at 900 W/m^2^ and 25 °C. Meanwhile, the maximum average difference is 2.18% (PV module Shell ST40). The minimum average error is 0.07% (PV module MSX-60).

In conclusion, the I_sc_ and V_oc_ values of PV modules under different operating conditions can be measured using the linear extrapolation method with an average error of less than 2.2% for V_oc_ and 1.2% for I_sc_.

### 3.6. Identifying the Potential MPP

To implement the proposed method, the I(V) characteristic curves of PV modules are investigated to determine the linear region as part of the study. After that, the I_sc_ and V_oc_ are determined based on the d value of the DC/DC converter by extrapolating from two random points in this limit. This solution not only reduces the disconnection time between the source and the load but also increases the calculation accuracy of I_sc_ and V_oc_ compared to updating them periodically. Finally, the Perturbation and Observation algorithm will verify the calculated MPP coordinates based on FF [18,28]. It ensures that the optimal operating point of the PV system is determined correctly if there is an error in the calculation of I_sc_ and V_oc_. The detailed sequence of steps is as follows:Step 1. Calculate I_sc_ and I_mp_: Select the values of d_1_, d_2_, d_3_, and d_4_ to calculate I_sc_ according to (23) and calculate the I_mp_ according to (10), respectively.Step 2. Calculate V_oc_ and V_mp_: Determine V_oc_ according to (27) and calculate the V_mp_ according to (10), respectively.Step 3. Estimate P_mp_: Calculate the potential P_mp_ of the PV system based on I_sc_ and V_oc_ according to (29).
(29)$$P_{mp}=I_{mp}V_{mp}=k_{v}V_{oc}k_{i}I_{sc}$$

Step 4. Calculate d_mp_: Calculate d_mp_ at potential MPP according to (9), which is later chosen as the starting reference point for the P&O method to find the actual duty cycle at the optimal MPP (d_opt_). It is the obtained duty cycle value when the solution converges.Step 5. Calculate the power at the optimal MPP (P_opt_): Change the Δd value to observe the output power and voltage. Then, compare them to the respective reference values to locate the optimal MPP. The algorithm converges when Equation (30) is satisfied.


(30)
$$\left|\Delta P\right|=\frac{P_{i+1}-P_{i}}{P_{i}}\times 100\%\leq \varepsilon$$


If the constraint is not satisfied, the algorithm double-checks the voltage error ΔV according to (31) to adjust accordingly.
(31)$$\Delta V=\frac{V_{i+1}-V_{i}}{V_{i}}\times 100\%$$

If ΔV × ΔP > 0, reduce d to increase V; if ΔV × ΔP < 0, increase d to reduce V.

Conversely, if expression (30) is satisfied, the algorithm converges and simultaneously checks the current deviation between two consecutive measurements to detect sudden changes in operating conditions according to (32). If the working environment is stable, the algorithm will continue to check the power deviation according to (30). Otherwise, the system restarts from the beginning.
(32)$$\left|\Delta I\right|=\frac{I_{i+1}-I_{i}}{I_{i}}\times 100\%\leq \varepsilon$$

As a result, Figure 8 presents the flowchart of the proposed method.

## 4. Results and Discussion

The proposed solution is simulated and experimented with, evaluating its effectiveness in the following situations:Compare the d_mp_ value with the d_opt_ to highlight the effectiveness of quickly calculating the starting value compared to scanning the entire I(V) curve. Check the error power (ΔP%) between the initially estimated power (P_mp_) and the P_max_ of each case. Further, the optimal convergence time (T_opt_) and simulation efficiency (η_opt_) between P_opt_ and P_max_ are also collected under different radiation and temperature conditions.Compare the T_opt_ and η_opt_ of the proposed solution with those of the traditional P&O [29], the VSSP&O [18], and the PSO [13] in the same operating conditions. This work demonstrates that the starting point of the proposed method plays an essential role in improving the speed and performance of the algorithm.Combine the Boost converter with the Chroma 62050H-600S PV simulator, which meets the PV parameters shown in Table 1, to experiment under the recommended operating conditions. The comparison results of the experiment convergence time (T_e_) and MPPT efficiency (η_e_) between output power (P_e_) and P_max_ are also investigated and collected.

To collect evaluation data for the above proposals, simulation and test scenarios under different operating conditions are proposed in Table 9. Further, additional simulation and experiment cases are also tested, such as radiation and temperature increase or decrease together, or one parameter increasing and the other decreasing. This evaluates the proposed solution’s response speed when working conditions change suddenly. Figure 9 shows the structure diagram of the proposed method in the PSIM environment.

### 4.1. Evaluate the Dmp and Pmp Values of the Proposed Algorithm

Figure 10 shows the output parameters under standard conditions, while Figure 11 compares the deviance between d_mp_ and d_opt_ in all proposed working conditions. The data pointed out that.

The calculated d_mp_ value at standard condition is 0.63 compared with 0.64 of the d_opt_. Therefore, the solution only undergoes 5 adjustment steps to converge at d_opt_. Further, the current and voltage waveforms are also stable at 0.0175 s after the d_opt_ value stops at 0.0168 s (Figure 10). The estimated power at d_mp_ is P_mp_ = 237.04 W compared to the P_max_ = 240.20 W, reaching 98.68%. The most significant calculation error is the No.1 because d_mp_ = 0.25 compared to d_opt_ = 0.3. Therefore, it undergoes eight adjustment steps to reach the convergence value. The most minor calculation error is in No.10. It takes only two adjustment steps to stabilize the system. This is also the case with the most significant power error (ΔP% = 2.9%). The minimum error is 1.03% (No.8). The average deviation for all tested cases is 1.83% (Table 10). The results show that d_mp_ always approximates the d_opt_ of the PV system under all proposed simulation conditions. Therefore, the adjustment iterations can be significantly reduced to reach the convergence value.

### 4.2. Evaluate the Convergence Speed and Performance of the Proposed Algorithm

The output power under standard conditions is P_opt_ = 240.19 W compared to P_max_ = 240.20 W (MPPT efficiency is about η_opt_ = 99.99%). This is also the maximum performance, and that of the minimum is 99.54% (No.1). The output powers at the optimal MPP are always approximately P_max_, the average efficiency being over 99.86% (Table 10).

Because the potential MPP is determined quickly, the solution has significantly reduced the number of iterations to increase the convergence speed to the optimal MPP. Specifically, the fastest speed is 0.01 s (No.10) due to only two adjustment steps. The slowest time is 0.022 s (No.1) with eight iterations, and the average speed is 0.018 s (Table 10).

Further, the proposed method’s MPPT performance, convergence speed, and iterations are compared with classic P&O, VSSP&O, and PSO algorithms in the same operating conditions and initial starting position. That means the starting value of P&O and VSSP&O is d = 0.1, and that of the proposed solution is from the lowest (d = 0.05) to the highest (d = 0.8). Because it has to calculate starting point (d_mp_) according to (9), this parameter ensures objectivity when comparing the convergence speed and performance of solutions. The adjustment step size of VSSP&O is Δd = 0.4(dP/dV). It allows the duty cycle’s step size to adjust automatically without setting the maximum and minimum values [12]. It is explained that if the initial searching point is far from the actual MPP location, the deviation dP/dV is large, so Δd increases. In contrast, when the searching position is close to MPP, this value is small, and Δd is automatically adjusted to decrease. The correction factor is 0.4 to ensure minimal error at the stable position. Because of the significant step size, converging at the MPP point will be challenging. Conversely, a small step size increases the search time. The step size for the remaining two solutions is Δd = 0.015. Finally, the main parameters of the PSO algorithm include the population size (N = 3), inertia weight (w = 0.25), cognitive coefficient (c_1_ = 0.02), and social coefficient (c_2_ = 0.5). In summary, the results show that:

Under standard operating conditions, the proposed solution only needs five adjustment steps from d_mp_ to converge. Meanwhile, the traditional P&O requires 36 iterations to scan the entire P(V) curve. Although the step size is adjusted, the VSSP&O also needs 16 adjustment steps to achieve the optimal MPP. With five adjustment steps for each individual, the PSO algorithm needs 15 calculations for all populations (Figure 12). In this case, the proposed solution has reduced the number of iterations compared to traditional P&O, VSSP&O, and PSO by 86%, 69%, and 67%, respectively. Therefore, the proposed method has the fastest convergence speed of 0.019 s. The traditional P&O takes 0.090 s, the VSSP&O is 0.020 s, and the PSO needs 0.037 s to reach the optimal position. This outstanding advantage helps the proposed solution get a fast convergence speed and is far ahead of the remaining algorithms.

The combined results in Table 11 show that the slowest convergence rate is the P&O algorithm. Which usually has a convergence time of more than 0.09 s. Its fastest speed in case of 10 is also 0.037 s. In contrast, the proposed solution is consistently outperforming in search speed. Although the convergence speed of VSSP&O and PSO is faster than that of P&O, it is still slower than that of the proposed solution in all survey cases. The average convergence speed of the proposed solution is 0.017 s. In contrast, that of P&O, VSSP&O, and PSO are 0.079 s, 0.022 s, and 0.38 s, respectively. Although the search times of the solutions are different, their MPPT performance is not significantly different. The average efficiency of MPO, P&O, VSSP&O, and PSO are 99.89%, 98.04%, 98.68%, and 99.26%, respectively (Table 11). The results show that the proposed solution always has better performance and convergence speed than the remaining solutions.

Another scenario is proposed to simulate and compare the dynamic response between the proposed method and other algorithms. Assuming the system is working stably, the operating conditions change suddenly, and the proposed solution needs to redefine the parameters I_sc_ and V_oc_ to find d_mp_ again. During the survey and research, the authors found that the current value is affected more than the voltage value under changing operating conditions. Therefore, this study uses expression (32) to detect sudden changes in radiation and temperature. If the current difference between two consecutive measurements is insignificant, the system performs P&O iterations around the operating point. However, if there is a significant current error, the solution will immediately reset the search method from the first step. The simulation results are presented in Figure 13, Figure 14, Figure 15 and Figure 16, while Figure 17, Figure 18, Figure 19, Figure 20, Figure 21, Figure 22, Figure 23 and Figure 24 show the experimental scheme and the results obtained under the proposed operating conditions.

First, the system operates stably with a sudden increase in radiation and temperature at 0.20 s (Figure 13). The results show that P&O has the worst response when it only reaches 188.15 W (about 93.41%) compared to the maximum power of 201.42 W at 0.091 s. Further, VSSP&O can improve the speed significantly (0.025 s), but the efficiency is only 97.34% (about 196.06 W). The proposed solution responds the fastest to fluctuations in operating conditions and reaches approximately 99.94% at 0.017 s. While the PSO algorithm’s efficiency is 99.24%, and the convergence time is 0.037 s.

Second, when there is a sudden decrease in radiation and temperature in the system, the MPPT capabilities of the solutions are presented in Figure 14. The convergence speed of the proposed solution is an outstanding advantage. As soon as the operating condition changed at 0.20 s, it converged after 0.019 s. Meanwhile, P&O, PSO, and VSSP&O spent 0.080 s, 0.037 s, and 0.021 s, respectively. Although their performance is similar, the power and duty cycle waveforms of the traditional P&O algorithm are less stable than other solutions.

Third, the system works under stable conditions with dropped irradiation but increases temperature immediately at 0.2 s (Figure 15). The proposed solution responds fastest when converging at 0.015 s, and VSSP&O stops at 0.025 s. The P&O method has the slowest convergence speed of 0.086 s. While the PSO algorithm needs 0.0371 s. The efficiency of the solutions is the same at over 99%.

Finally, the system operates stably, then the radiation increases, and the temperature decreases at 0.2 s (Figure 16). The comparison results show that P&O responds the worst. Its convergence speed is 0.091 s compared to 0.017 s of the proposed method, 0.018 s of the VSSP&O, and 0.0375 s of the PSO.

In conclusion, the proposed solution always has a superior convergence speed compared to the remaining methods in the testing conditions. This advantage is due to an accurate prediction of the d_mp_ value. Furthermore, changing operating conditions makes the convergence speed less affected. Because the start point of the search loops (d_mp_) always approximates the optimal value (d_opt_). The average MPPT speed is about 0.017 s. Although VSSP&O has step size adjustment to increase MPPT speed, the average time is about 0.022 s. Further, the search time of the PSO algorithm is about 0.038 s. Finally, the traditional P&O method needs more time when the average convergence speed is about 0.079 s. The testing results show that the proposed solution has a high MPPT efficiency, stability in continuously changing operating conditions, and superior convergence speed compared to other algorithms.

### 4.3. Experiment with the Proposed Algorithm

The proposed solution has been experimented with on a PV Chroma 62050H-600S simulator (Figure 17). It is connected to the resistive load by a boost converter. The main parameters of the DC/DC converter can be calculated according to reference [30] and are listed in Table 12. The system is tested based on the operating conditions as previously simulated. The results under conditions such as attenuated radiation (Figure 18) and increased temperature (Figure 19), and both parameters differ from standard conditions (Figure 20), showing the most excellent MPPT efficiency can reach 99.38%. The lowest efficiency value is 92.87%, and the average efficiency value is over 96.56%. The fastest MPPT speed is 0.03 s, the slowest is 0.31 s, and the average time is about 0.14 s. Further, the duty cycle ranges from 0.26 to 0.74 when the operating environment changes with a wide fluctuation (Table 13). It is asymptotic from both sides of d = 0.5 to ensure there is not too much pressure on the switches compared to working at extremes.

**Table 12 sensors-23-07503-t012:** Specification of the boost converter.

Parameters	Value
Input voltage	20 V
Output voltage	70 V
Output power	300 W
Ripple voltage	5%
Ripple current	5%
Electrolytic capacitor C_in_	1000 μF
Electrolytic capacitor C_out_	100 μF
Inductor L	0.17 mH
Diode D	MUR3060PT
MOSFET	FDA50N50
Switching frequency	40 kHz
Sampling time	1.5 μs

**Table 13 sensors-23-07503-t013:** The experimental MPPT performance.

No.	Output Power P_e_ (W)	Experimental Performance η_e_(%)	Duty Cycle	Convergence Speed T_e_ (s)
1	59.61	99.38	0.35	0.05
2	117.8	98.24	0.55	0.05
3	169.2	94.02	0.64	0.06
4	226.7	94.5	0.7	0.14
5	237.3	96.9	0.71	0.31
6	225	95.76	0.72	0.1
7	213.6	94.95	0.73	0.15
8	207	96.26	0.73	0.35
9	190.4	92.87	0.74	0.15
10	48.27	98.75	0.26	0.12
11	68.88	97.74	0.42	0.12
12	90.59	98.49	0.47	0.21
13	126.4	95.75	0.62	0.16
14	163.4	97.21	0.7	0.15
15	175.9	97.66	0.72	0.03

**Figure 17 sensors-23-07503-f017:**
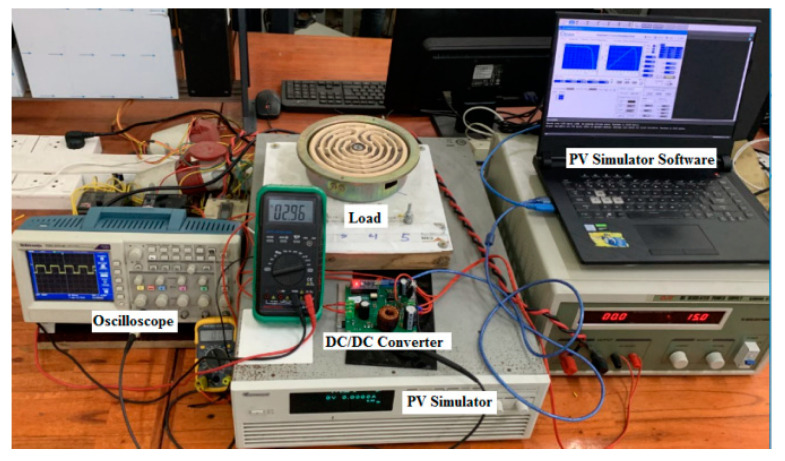
Experimental system setup.

Meanwhile, Table 14 shows that the proposed solution has superior convergence speed and performance compared to the latest studies. Its respective values are 0.010 s and 99.99%, respectively. Hybrid solutions can significantly reduce the search time, 0.016 s for the COA-FLC algorithm, but the performance is low, 99.83%. In contrast, intelligent algorithms can improve the performance to 99.99% (GWO), but its disadvantage is slow convergence, 0.190 s.

**Table 14 sensors-23-07503-t014:** Comparison of the algorithms.

Algorithms	Convergence Speed (s)	MPPT Efficiency (%)
MPO	0.01	99.99
PSO+INC [11]	0.0434	99.40
PSO+P&O [11]	0.0495	99.00
LBNS [12]	0.038	99.98
CSA [13]	0.48	99.90
GWO [13]	0.19	99.99
MIC [13]	0.14	99.90
PSO [13]	0.92	99.96
MC-P&O [14]	0.0375	99.54
BMCA [16]	0.096	96.70
COA-FLC [17]	0.016	99.83

**Figure 18 sensors-23-07503-f018:**
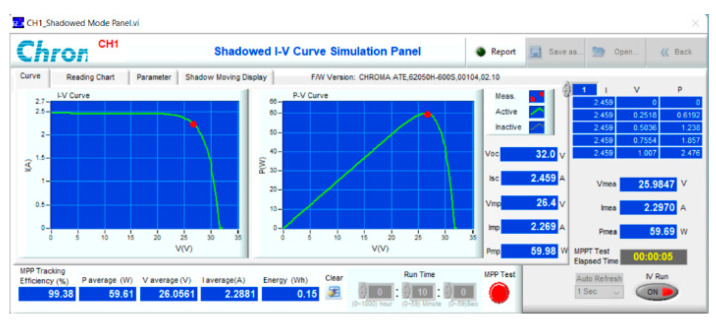
Experiment MPPT for No. 01.

**Figure 19 sensors-23-07503-f019:**
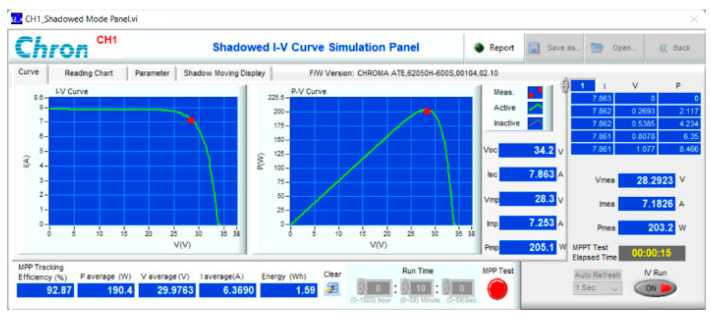
Experiment MPPT for No. 09.

**Figure 20 sensors-23-07503-f020:**
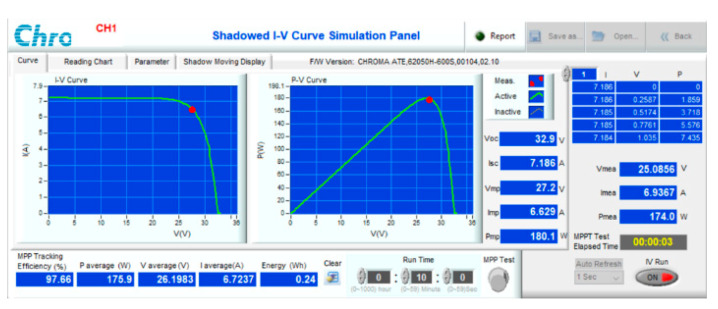
Experiment MPPT for No. 15.

To test the dynamic response of the proposed solution when operating conditions change continuously, experimental scenarios are deployed, including a sudden increase in radiation and temperature (Figure 21), a decrease in radiation and temperature (Figure 22), a reduction in radiation but increase in working temperature (Figure 23), and finally increase in radiation but decrease in temperature (Figure 24). The time axis from 0 to 100 in Figure 21, Figure 22, Figure 23 and Figure 24 represents a total simulation time of 20 s. While the system operates stably, the temperature and radiation parameters suddenly change at 10 s. The power waveform shows that when the radiation increases, the new MPP search time is about 1.0 s (Figure 21), and the efficiency is 94.68%. Meanwhile, stable speed and MPPT performance increase significantly in cases of radiation reduction. After only 0.7 s, the system stabilized and achieved an efficiency of 99.58% (Figure 22).

**Figure 21 sensors-23-07503-f021:**
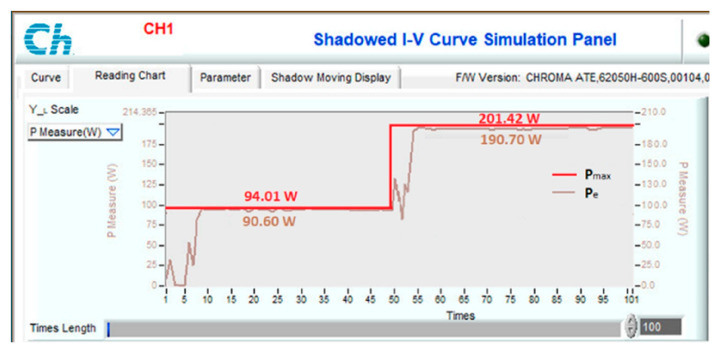
MPPT under the operating conditions from No. 12 to No. 09.

**Figure 22 sensors-23-07503-f022:**
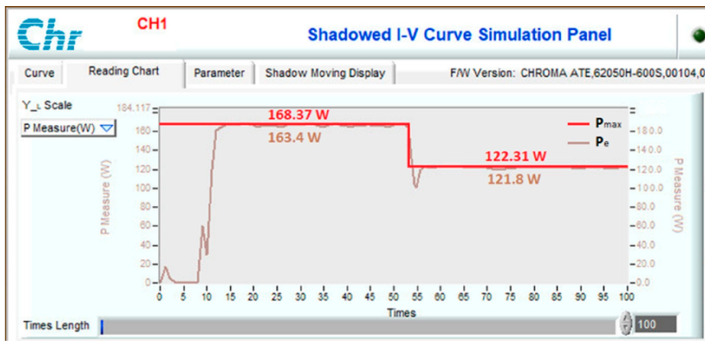
MPPT under the operating conditions from No. 14 to No. 02.

**Figure 23 sensors-23-07503-f023:**
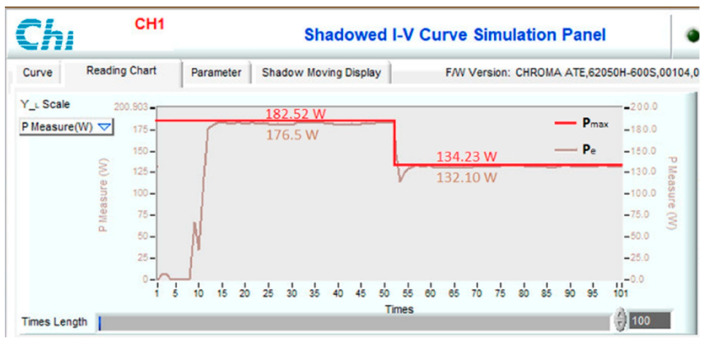
The power waveform under experimental conditions varies from No. 03 to No. 13.

**Figure 24 sensors-23-07503-f024:**
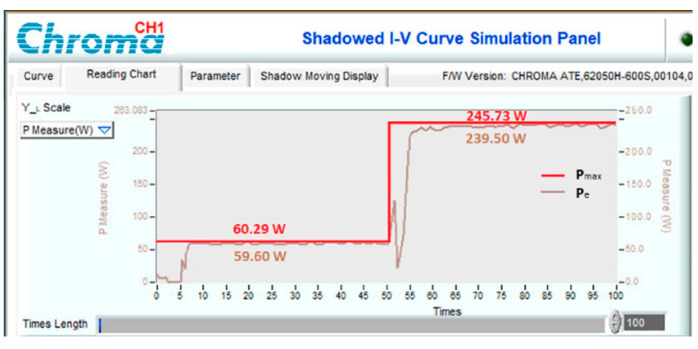
The power waveform under experimental conditions varies from No. 01 to No. 05.

The above results and discussions show that the proposed solution promises to bring convenient applications in a parallel configuration, where medium and small capacity power is used.

## 5. Conclusions

The paper presents an MPPT method for the photovoltaic system based on an MPO algorithm. The V_oc_ and I_sc_ are calculated directly based on the d value to quickly determine the starting point for the proposed solution. A comprehensive set of studies was conducted under standard test conditions and variations in irradiance and temperature. These studies aimed to thoroughly examine the system’s performance under different scenarios. Both simulation results in the PSIM environment and experiment results on the Chroma model show that the proposed method’s MPPT speed is always superior to other algorithms in the same testing conditions. It also has a faster dynamic response and more stability when PV systems operate in changing conditions (about 0.01 s). Further, the quick determination of the potential MPP value significantly limits the search space, reduces the computational burden, and improves performance. The comparison results show that the MPO solution has the highest dynamic rendition (approximately 100%). It has a high potential to be widely and reliably applied in applications with low voltage and power requirements. In addition, the article also presents the survey data of the I(V) characteristic curves of some typical PV modules, which can be used as a reference for further research on photovoltaic systems. The limitation of this study is that the proposed solution only applies to the parallel configuration of PV panels, which has only one extreme in all operating conditions. Therefore, the applied research to determine the global maximum power point in partial shade conditions with many local extremes will be the goal of the following studies by the authors.

## Figures and Tables

**Figure 1 sensors-23-07503-f001:**
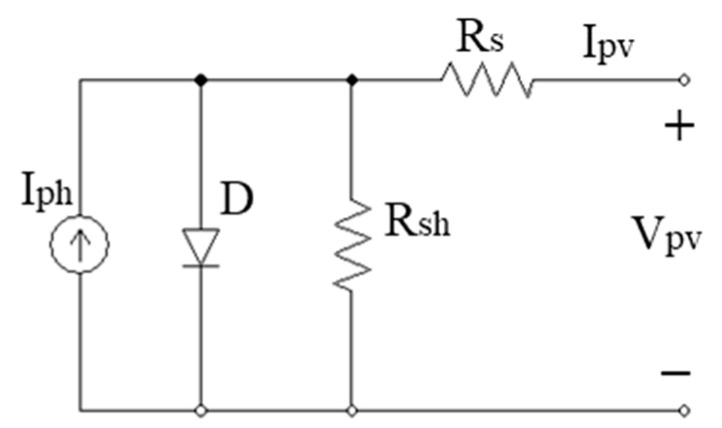
Equivalent circuit of PV cell.

**Figure 2 sensors-23-07503-f002:**
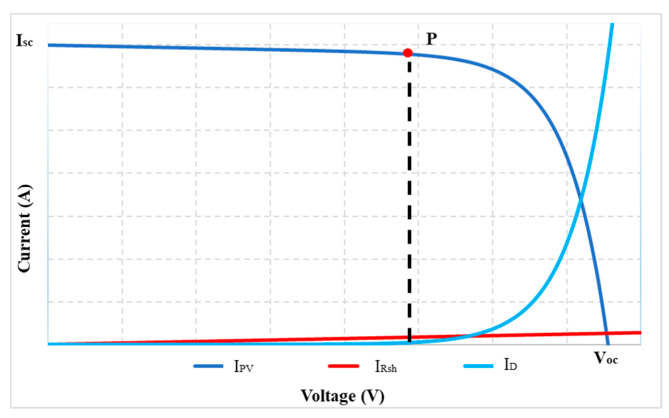
The I(V) characteristics under standard working conditions.

**Figure 3 sensors-23-07503-f003:**
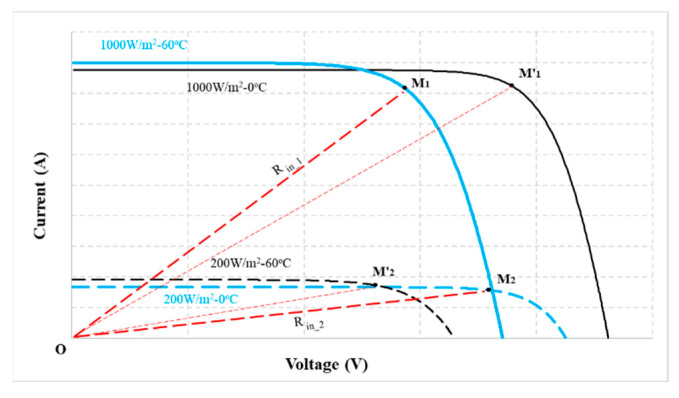
The I(V) characteristics under different working conditions.

**Figure 4 sensors-23-07503-f004:**
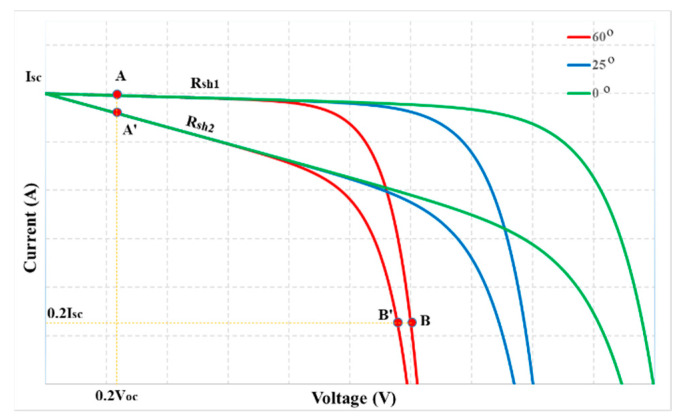
The I(V) characteristics under different conditions of temperature and R_sh_.

**Figure 5 sensors-23-07503-f005:**
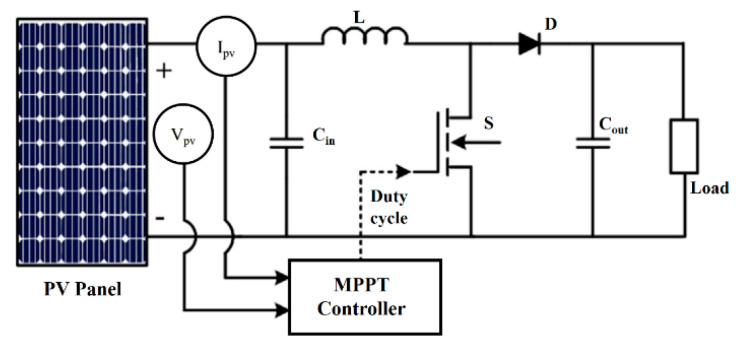
Schematic diagram of the PV control system in this research.

**Figure 6 sensors-23-07503-f006:**
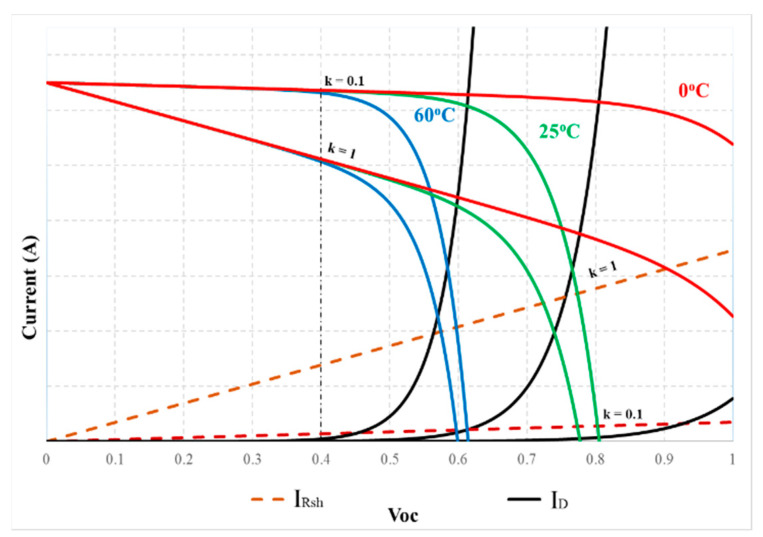
The I(V) characteristic curve of the PV module MSX-60.

**Figure 8 sensors-23-07503-f008:**
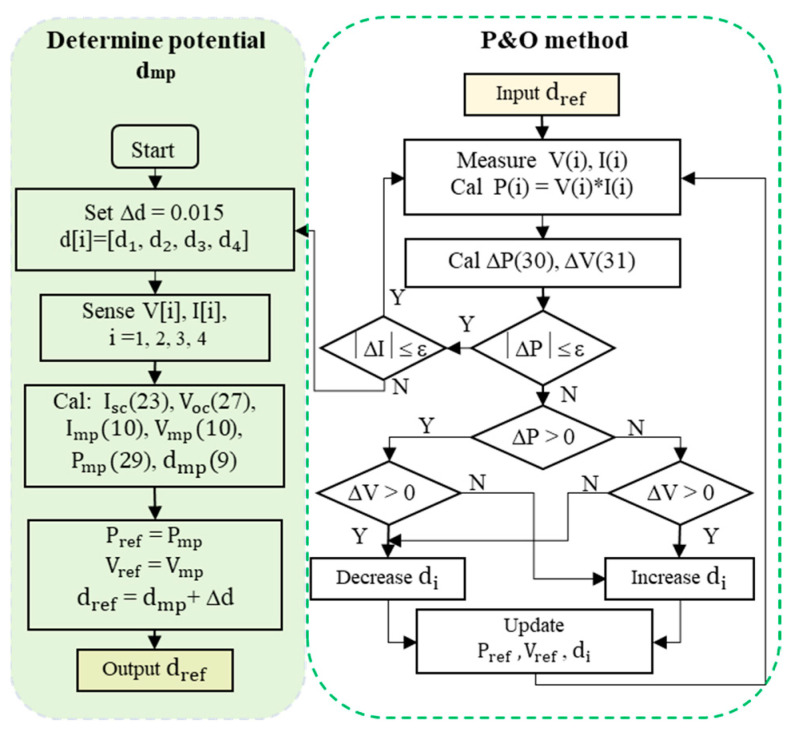
Algorithm flowchart of the proposed method.

**Figure 9 sensors-23-07503-f009:**
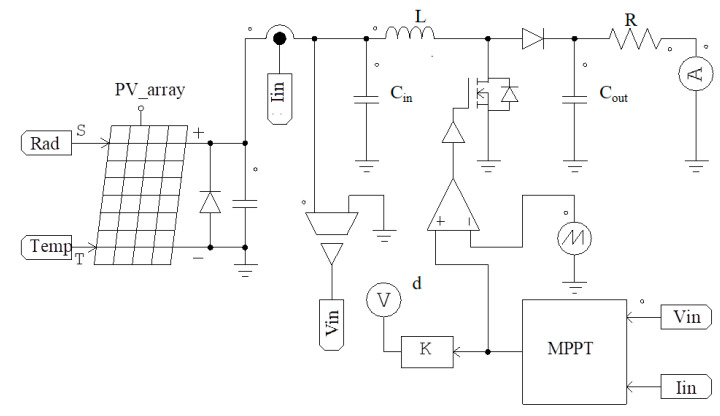
Schematic diagram of the proposed method.

**Figure 10 sensors-23-07503-f010:**
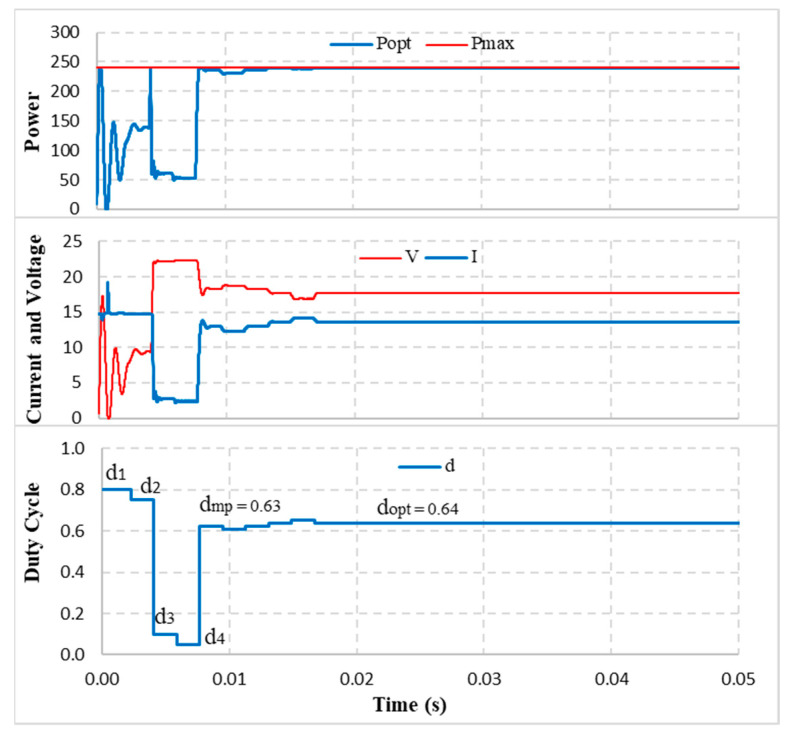
MPPT simulation under standard condition.

**Figure 11 sensors-23-07503-f011:**
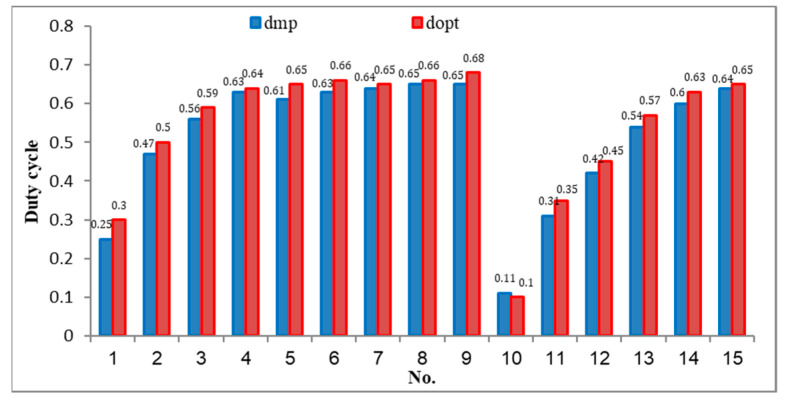
Compare the d_mp_ values with d_opt_.

**Figure 12 sensors-23-07503-f012:**
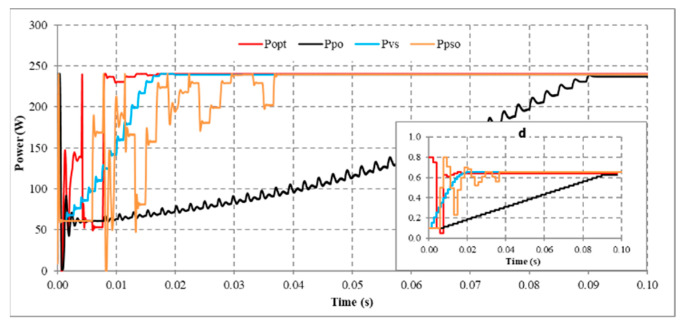
Output waveform under standard operating conditions (No.4).

**Figure 13 sensors-23-07503-f013:**
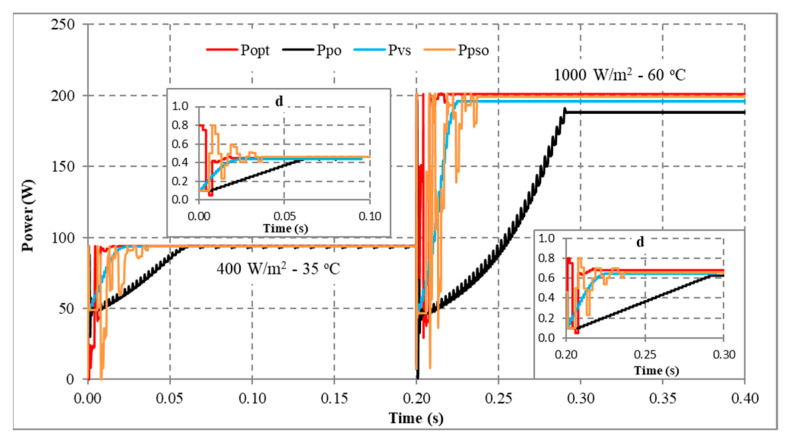
MPPT when solar irradiation and temperature increased.

**Figure 14 sensors-23-07503-f014:**
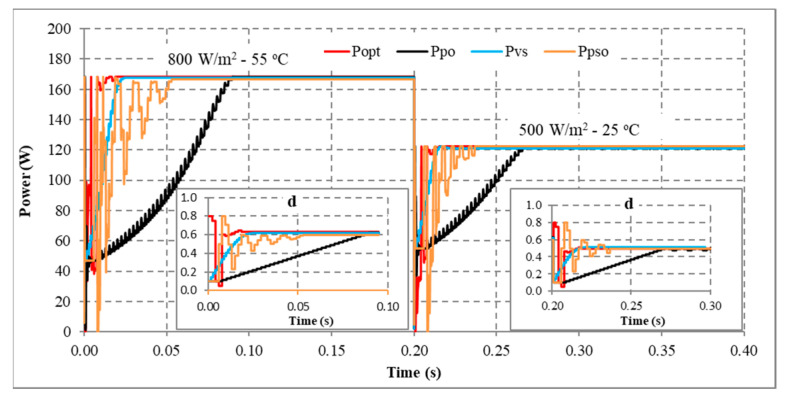
MPPT when solar irradiation and temperature decreased.

**Figure 15 sensors-23-07503-f015:**
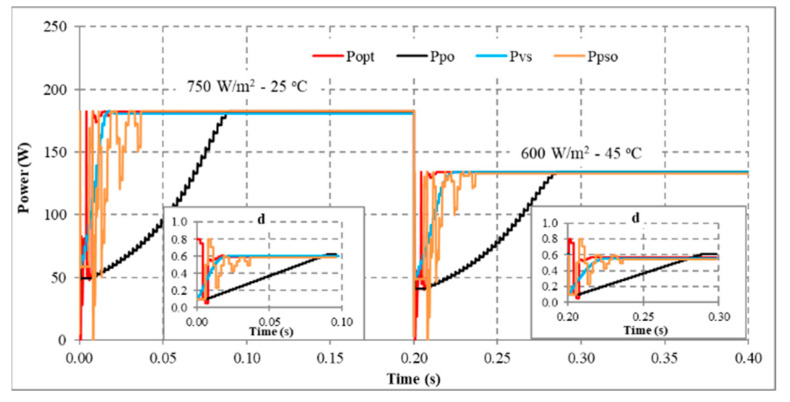
MPPT under the operating conditions from No. 03 to No. 13.

**Figure 16 sensors-23-07503-f016:**
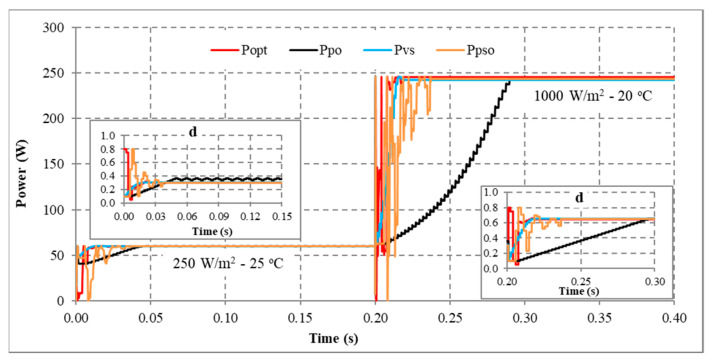
MPPT under the operating conditions from No. 01 to No. 05.

**Table 1 sensors-23-07503-t001:** Specifications of the PV panel MSX-60.

Parameters	Value
Number of cells	36
Series Resistance	0.2 (Ω)
Shunt Resistance	304.83 (Ω)
Short Circuit Current	3.8 (A)
Open Circuit Voltage	21.1 (V)
Maximum Power Point Voltage	17.1 (V)
Maximum Power Point Current	3.5 (A)
The diode reverse saturation current	9.094 × 10^−8^ (A)
Temperature coefficient of I_sc_	3 (mA/°C)
Temperature coefficient on V_oc_	−80 (mV/°C)

**Table 2 sensors-23-07503-t002:** The best region of d.

Type of PV Module	d_mp1_	d_mp2_
MSC-60	0.86	0.80
Shell SP75	0.82	0.75
Shell SQ150	0.84	0.78
SSt 230-60P	0.83	0.77
Shell S70	0.83	0.75
GxB-340	0.85	0.79
Shell ST40	0.81	0.72

**Table 3 sensors-23-07503-t003:** The duty cycle d_A_ at 0.2 V_oc_ and 0.4 V_oc_.

Type of PV Module	0.2 V_oc_	0.4 V_oc_
MSX-60	0.29	0.71
Shell SP75	0.36	0.64
Shell SQ150	0.33	0.67
SSt 230-60P	0.34	0.66
Shell S70	0.36	0.64
GxB-340	0.31	0.69
Shell ST40	0.41	0.60

**Table 4 sensors-23-07503-t004:** The ΔI% under different radiations.

Type of PV Module		Radiation (W/m^2^)						
		200	350	500	650	800	900	1000
MSX-60	I_sc_	1.48	2.59	3.7	4.81	5.92	6.66	7.41
	I_sc_cal_	1.5	2.63	3.75	4.89	6.01	6.76	7.51
	ΔI%	1.35	1.54	1.35	1.66	1.52	1.5	1.35
Shell SP75	I_sc_	1.89	3.32	4.74	6.17	7.6	8.56	9.4
	I_sc_cal_	1.9	3.33	4.75	6.18	7.6	8.55	9.5
	ΔI%	0.53	0.3	0.21	0.16	0	0.12	1.06
Shell SQ150	I_sc_	1.89	3.32	4.73	6.16	7.58	8.54	10
	I_sc_cal_	1.89	3.32	4.74	6.16	7.58	8.53	9.9
	ΔI%	0	0	0.21	0	0	0.12	1
SSt 230-60P	I_sc_	3.36	5.88	8.41	10.93	13.45	15.14	16.83
	I_sc_cal_	3.36	5.88	8.41	10.94	13.45	15.13	16.81
	ΔI%	0	0	0	0.09	0	0.07	0.12
Shell S70	I_sc_	1.77	3.1	4.43	5.76	7.08	7.97	8.86
	I_sc_cal_	1.77	3.12	4.42	5.76	7.08	7.97	8.85
	ΔI%	0	0.65	0.23	0	0	0	0.11
GxB-340	I_sc_	3.66	6.41	9.15	11.91	14.65	16.49	18.33
	I_sc_cal_	3.66	6.41	9.15	11.91	14.6	16.48	18.3
	ΔI%	0	0	0	0	0.34	0.06	0.16
Shell ST40	I_sc_	1.05	1.85	2.63	3.43	4.22	4.76	5.32
	I_sc_cal_	1.05	1.85	2.64	3.43	4.22	4.75	5.28
	ΔI%	0	0	0.38	0	0	0.21	0.75

**Table 5 sensors-23-07503-t005:** The ΔI% under different temperatures.

Type of PV Module		Temperature (°C)						
		0	10	20	30	40	50	60
MSX-60	I_sc_	7.28	7.33	7.39	7.47	7.59	7.59	7.6
	I_sc_cal_	7.39	7.44	7.49	7.54	7.59	7.64	7.69
	ΔI%	1.51	1.5	1.35	0.94	0	0.66	1.18
Shell SP75	I_sc_	9.35	9.41	9.48	9.57	9.68	9.85	10.1
	I_sc_cal_	9.35	9.41	9.47	9.53	9.58	9.65	9.75
	ΔI%	0	0	0.11	0.42	1.03	2.03	3.47
Shell SQ150	I_sc_	9.41	9.45	9.5	9.58	9.71	9.94	10.3
	I_sc_cal_	9.41	9.44	9.48	9.53	9.6	9.72	9.91
	ΔI%	0	0.11	0.21	0.52	1.13	2.21	3.79
SSt 230-60P	I_sc_	16.63	16.7	16.79	16.88	16.99	17.14	17.37
	I_sc_cal_	16.62	16.7	16.78	16.85	16.93	17.01	17.08
	ΔI%	0.06	0	0.06	0.18	0.35	0.76	1.67
Shell S70	I_sc_	8.76	8.8	8.84	8.88	8.93	9	9.08
	I_sc_cal_	8.75	8.79	8.83	8.87	8.91	8.95	8.99
	ΔI%	0.11	0.11	0.11	0.11	0.22	0.56	0.99
GxB-340	I_sc_	18.01	18.14	18.27	18.4	18.55	18.72	18.93
	I_sc_cal_	18.01	18.13	18.2	18.3	18.5	18.6	18.74
	ΔI%	0	0.06	0.38	0.54	0.27	0.64	1.00
Shell ST40	I_sc_	5.26	5.28	5.3	5.34	5.41	5.53	5.72
	I_sc_cal_	5.27	5.29	5.33	5.4	5.5	5.6	5.84
	ΔI%	0.19	0.19	0.57	1.12	1.66	1.27	2.1

**Table 6 sensors-23-07503-t006:** The d_B_ values at 0.2 I_sc_ under different conditions.

Type of PV Module	M_1_	M_2_
MSC-60	−0.71	0.3
Shell SP75	−0.54	0.13
Shell SQ150	−0.61	0.22
SSt 230-60P	−0.59	0.19
Shell S70	−0.54	0.14
GxB-340	−0.66	0.25
Shell ST40	−0.42	0.03

**Table 7 sensors-23-07503-t007:** The ΔV% under different radiations.

Type of PV Module		Radiation (W/m^2^)						
		200	350	500	650	800	900	1000
MSX-60	V_oc_	37.24	39.17	40.29	41.03	41.61	41.93	42.21
	V_oc_cal_	38.29	39.62	40.5	41.13	41.63	41.92	42.17
	ΔV%	2.82	1.15	0.52	0.24	0.05	0.02	0.09
Shell SP75	V_oc_	38.62	40.52	41.63	42.36	42.93	43.25	43.53
	V_oc_cal_	39.77	41.1	41.98	42.61	43.11	43.4	43.65
	ΔV%	2.98	1.43	0.84	0.59	0.42	0.35	0.28
Shell SQ150	V_oc_	76.06	80.82	82.99	84.43	85.57	86.2	86.75
	V_oc_cal_	79.09	81.75	83.51	84.76	85.87	86.34	86.85
	ΔV%	3.98	1.15	0.63	0.39	0.35	0.16	0.12
SSt 230-60P	V_oc_	65.05	68.29	70.17	71.41	72.38	72.91	73.38
	V_oc_cal_	66.98	69.19	70.67	71.71	72.56	73.03	73.45
	ΔV%	2.97	1.32	0.71	0.42	0.25	0.16	0.1
Shell S70	V_oc_	37.21	39.24	40.39	41.15	41.74	42.06	42.35
	V_oc_cal_	38.54	39.87	40.75	41.38	41.88	42.17	42.42
	ΔV%	3.57	1.61	0.89	0.56	0.34	0.26	0.17
GxB-340	V_oc_	90.52	95.18	97.81	99.53	100.88	102.62	102.8
	V_oc_cal_	93.75	96.81	98.85	100.3	101.47	102.12	102.71
	ΔV%	3.57	1.71	1.06	0.77	0.58	0.49	0.09
Shell ST40	V_oc_	40.76	42.88	44.01	44.75	45.33	45.65	45.93
	V_oc_cal_	42.67	44	44.88	45.51	46.01	46.3	46.55
	ΔV%	4.69	2.61	1.98	1.7	1.5	1.42	1.35

**Table 8 sensors-23-07503-t008:** The ΔV% under different temperatures.

Type of PV Module		Temperature (°C)						
		0	10	20	30	40	50	60
MSX-60	V_oc_	45.99	44.49	42.97	41.44	39.91	38.38	36.83
	V_oc_cal_	45.94	44.44	42.93	41.41	39.89	38.36	36.82
	ΔV%	0.11	0.11	0.09	0.07	0.05	0.05	0.03
Shell SP75	V_oc_	47.18	45.73	44.26	42.79	41.32	39.84	38.35
	V_oc_cal_	47.3	45.84	44.38	42.92	41.44	39.96	38.47
	ΔV%	0.25	0.24	0.27	0.3	0.29	0.3	0.31
Shell SQ150	V_oc_	94.15	91.2	88.24	85.27	82.28	79.27	76.26
	V_oc_cal_	94.22	91.28	88.33	85.37	82.38	79.39	76.38
	ΔV%	0.07	0.09	0.1	0.12	0.12	0.15	0.16
SSt 230-60P	V_oc_	79.44	77.02	74.6	72.16	69.71	67.25	64.78
	V_oc_cal_	79.49	77.08	74.66	72.24	69.79	67.34	64.87
	ΔV%	0.06	0.08	0.08	0.11	0.11	0.13	0.14
Shell S70	V_oc_	46.12	44.62	43.16	41.59	40.06	38.53	36.99
	V_oc_cal_	46.18	44.68	43.18	41.66	40.14	38.61	37.08
	ΔV%	0.13	0.13	0.05	0.17	0.2	0.21	0.24
GxB-340	V_oc_	108.45	105.99	103.51	101.02	98.52	95.99	93.46
	V_oc_cal_	108.87	106.42	103.95	101.46	98.96	96.44	93.91
	ΔV%	0.39	0.41	0.43	0.44	0.45	0.47	0.48
Shell ST40	V_oc_	49.45	48.09	46.73	45.35	43.97	42.58	41.14
	V_oc_cal_	50	48.63	47.26	45.88	44.48	43.09	41.68
	ΔV%	1.11	1.12	1.13	1.17	1.16	1.2	1.31

**Table 9 sensors-23-07503-t009:** The proposed simulation and experimental cases.

No.	Radiation (W/m^2^)	Temperature (°C)	P_max_ (W)
1	250	25	60.29
2	500	25	122.31
3	750	25	182.52
4	1000	25	240.2
5	1000	20	245.73
6	1000	30	234.77
7	1000	40	223.69
8	1000	50	212.57
9	1000	60	201.42
10	200	0	48.32
11	300	20	73.94
12	400	35	94.01
13	600	45	134.23
14	800	55	168.37
15	900	60	182.88

**Table 10 sensors-23-07503-t010:** Simulation results with the proposed solution.

No.	Number of Δd	P_mp_ (W)	ΔP%	P_opt_ (W)	η_opt_ (%)	T_opt_ (s)
1	8	58.78	2.5	60.01	99.54	0.022
2	6	119.9	1.97	122.3	99.99	0.019
3	6	178.91	1.98	182.39	99.93	0.019
4	5	237.04	1.32	240.19	99.99	0.019
5	6	239.98	2.34	245.45	99.89	0.019
6	6	229.91	2.07	234.15	99.74	0.019
7	5	219.85	1.72	223.46	99.9	0.017
8	5	210.39	1.03	212.5	99.97	0.017
9	6	197.66	1.87	201.29	99.94	0.019
10	2	46.92	2.9	48.28	99.91	0.01
11	5	72.79	1.69	73.7	99.68	0.017
12	6	92.28	1.81	94	99.99	0.018
13	4	132.78	1.08	134.18	99.96	0.015
14	6	164.61	2.12	168.27	99.94	0.019
15	5	180.94	1.06	182.86	99.99	0.016

**Table 11 sensors-23-07503-t011:** Comparison between MPPT algorithms.

No.	Convergence Speed (ms)				Efficiency (η%)			
	MPO	P&O	VSSP&O	PSO	MPO	P&O	VSSP&O	PSO
1	22	51	22	38.0	99.54	99.39	99.57	99.83
2	19	80	21	37.0	99.99	99.58	98.98	99.99
3	19	88	20	36.8	99.93	99.61	99.31	99.93
4	19	90	20	37.7	99.99	98.33	99.65	99.29
5	19	91	19	37.5	99.89	99.19	98.96	99.29
6	19	91	22	37.0	99.74	98.12	99.97	98.28
7	17	92	25	37.1	99.90	96.98	97.49	98.69
8	17	91	25	37.2	99.97	94.87	97.43	99.85
9	19	91	25	37.3	99.94	93.41	97.34	99.24
10	10	37	16	38.0	99.91	99.96	98.99	99.75
11	17	52	24	35.6	99.68	96.16	96.70	99.43
12	18	67	25	37.3	99.99	98.62	99.70	99.95
13	15	86	25	37.1	99.96	99.69	99.89	99.02
14	19	91	25	53.6	99.94	99.59	99.60	98.98
15	16	91	24	37.5	99.99	97.03	96.73	97.50

## Data Availability

The data presented in this study is available on request from the corresponding author.

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
