# Peer review of "Rapidly Determine the Maximum Power Point in the Parallel Configuration of the Photovoltaic System"

_sensors, 2023, doi:10.3390/s23177503_

Round 1
Reviewer 1 Report
This paper introduced a simple and fast convergence speed method of MPPT for PV generation system. The paper is well presented.
Two questions could be discussed as below.
1. In the section of 3, the predisposition of Formulation (9) is the load is a constant resistor. So is the MPPT method conducted in the paper suitable for other scenarios such as grid-tied PV generation or the load is not a constant resistor?
2. Does this method require higher consistency for PV panels of the same type? Is there any influence on the accuracy of the MPPT due to the deviation between different PV modules?
This paper is well organized and written.
Author Response
Thank you for allowing a resubmission of our manuscript, with an opportunity to address the reviewers’ comments.
Please see the attachment.
Best regards,
Van Hien Bui et al.

Reviewer 2 Report
The review report of manuscript # sensors-2553484
This paper presents a maximum power point tracker (MPPT) of photovoltaic energy systems to reduce the convergence time and enhance the efficiency of the tracker system.
The logic of the paper is not clear and many details are not shown in the manuscript. Moreover, the literature review on the techniques used to reduce the convergence time and failure rate are not shown and they are not discussed in the manuscript. The innovation and contribution of this research are not well presented. The following points should be considered:
1- The abstract is not acceptable at all, the abstract should define the problem, its importance, the existing solutions, the limitations of these solutions, and the innovation introduced in this paper to avoid these limitations.
2- The shading condition is mentioned several times but unfortunately, the authors did not consider it in the analysis or the results.
3- Many techniques have been used to reduce the convergence time shown in the literature such as the initialization of search agents at the predicted peaks of P-V curves, gradually reducing the number of search agents in modern metaheuristic optimization algorithms such as the musical chairs algorithm, scanning particle technique, etc., but the authors did not discuss any one of these important studies in the literature.
4- Many factors can affect the convergence time of the MPPT techniques such as the sampling time, switching frequency, and the design parameters of the DC/DC converter which are not included or discussed in the manuscript.
5- The results shown in Fig. 10 are not clear where is impossible to have these changes in the power and current in the first 0.005 s meanwhile the duty ratio is constant !!!!
6- There is no clear definition in the manuscript for the difference between dopt and dmp.
7- The authors did not show how they configure the sudden change in power due to shading change and what they should do in the initial values of search agents after this sudden change.
Author Response
Thank you for allowing a resubmission of our manuscript, with an opportunity to address the reviewers’ comments.
Please see the attachment
Best regards,
Van Hien Bui et al.

Reviewer 3 Report
1/ The novelty and objectives of this work should be clarified and described in the last paragraph of the introduction.
2/ The "Related Works" should be inserted separately with the Introduction section to be more complete. Then, in relation to these related works, you can describe the novelty of your work.
3/How does the boost circuit and proposed system used in the text achieve maximum power generation from the PV field? Please explain in detail.
4/ In the paper, only the circuits and models for the various components are presented. Please provide more practical scenarios and combine them with the models and algorithms proposed in the paper.
5/Which statistical criterium have you used to conclude about the performance of proposed method?
6/How does the boost circuit and proposed system used in the text achieve maximum power generation from the PV field? Please explain in detail.
7/ Add Future work in the section of conclusion
Author Response

(The authors gave the same response as above.)

Round 2
Reviewer 2 Report
The authors covered all comments in my previous report and the paper can be accepted in its present form.
Reviewer 3 Report
I have no other comment, just to improve the presentation quality of your article